# Effect of Hepatitis Viruses on the Nrf2/Keap1-Signaling Pathway and Its Impact on Viral Replication and Pathogenesis

**DOI:** 10.3390/ijms20184659

**Published:** 2019-09-19

**Authors:** Daniela Bender, Eberhard Hildt

**Affiliations:** Department of Virology, Paul-Ehrlich-Institut, Paul-Ehrlich-Straβe 51–59, D-63225 Langen, Germany; Daniela.Bender@pei.de

**Keywords:** Hepatitis C virus, Hepatits B virus, Nrf2, reactive oxygen species, liver regeneration

## Abstract

With respect to their genome and their structure, the human hepatitis B virus (HBV) and hepatitis C virus (HCV) are complete different viruses. However, both viruses can cause an acute and chronic infection of the liver that is associated with liver inflammation (hepatitis). For both viruses chronic infection can lead to fibrosis, cirrhosis and hepatocellular carcinoma (HCC). Reactive oxygen species (ROS) play a central role in a variety of chronic inflammatory diseases. In light of this, this review summarizes the impact of both viruses on ROS-generating and ROS-inactivating mechanisms. The focus is on the effect of both viruses on the transcription factor Nrf2 (nuclear factor erythroid 2 (NF-E2)-related factor 2). By binding to its target sequence, the antioxidant response element (ARE), Nrf2 triggers the expression of a variety of cytoprotective genes including ROS-detoxifying enzymes. The review summarizes the literature about the pathways for the modulation of Nrf2 that are deregulated by HBV and HCV and describes the impact of Nrf2 deregulation on the viral life cycle of the respective viruses and the virus-associated pathogenesis.

## 1. Introduction

Cells harbor effective antioxidant, detoxifying and cytoprotective mechanisms to maintain cellular homeostasis. One key factor regulating electrophilic and oxidative stress is Nrf2 (nuclear factor erythroid 2 (NF-E2)-related factor 2) that is ubiquitously expressed in many tissues and cell types [1] and is involved in the expression of 1055 target genes [2]. Thus, besides its role in cellular detoxification, Nrf2 has been described as being involved in various cellular processes including proliferation and differentiation [3,4,5], iron homeostasis [6], immune response [7], cell signaling, metabolism [5,7], cell cycle, cell survival [2], wound repair [8], liver regeneration [9,10], cancer and drug resistance [11,12,13,14].

Chronic infection by hepatitis B virus (HBV) and hepatitis C virus (HCV) leads to a chronic inflammation of the liver (hepatitis). For many chronic inflammatory diseases a deregulation of the intracellular level of reactive oxygen species (ROS) has been described. This review describes the effect of HBV and HCV on ROS-generating and -inactivating systems with a focus on the interference of these viruses with the Nrf2/ARE-dependent gene expression. The relevance for the viral life cycle and virus-associated pathogenesis is summarized.

## 2. The Hepatitis C Virus (HCV)

The hepatitis C virus (HCV) is a major cause of chronic liver diseases worldwide, including chronic hepatitis, fibrosis, cirrhosis and hepatocellular carcinoma (HCC). According to the World Health Organization (WHO), an estimated 71 million people are persistently infected with HCV [15]. The improvement of therapeutic approaches with new oral direct-acting antivirals (DAAs) enables 95% of infected individuals to be cured, reducing the risk of HCV-associated liver diseases. However, due to the high costs of these therapies, the limited availability and the lack of a protective vaccine, HCV will remain a global health burden [16].

HCV is an enveloped virus with a positive-orientated RNA genome that belongs to the *Hepacivirus* within the Flaviviridae family. The HCV virion displays an icosahedral structure with an average diameter of 50–80 nm [17]. The association with neutral lipids (triglycerides, cholesterol and cholesterol ester) and host lipoproteins (apoB, apoE, apoC1, C2, C3) termed these hybrid particles as lipoviroparticles (LVPs). In line with this, LVPs exhibit a low density in a range between 1.03 g/cm^3^ to 1.20 g/cm^3^, with low-density fractions being more infectious [18,19,20,21,22]. The host-derived, lipid-bilayer envelope harbors the viral glycoproteins E1 and E2 and surrounds the nucleocapsid that is composed of the homo-oligomerized core proteins associated with the viral RNA [23]. The viral RNA genome has a size of 9.6 kb and is flanked by high structured untranslated regions (UTRs). The IRES (internal ribosomal entry site)-mediated translation yields a polyprotein precursor of approximately 3010 amino acids (aa) that is co- and/or post-translationally processed by viral or cellular proteases into 10 mature polyproteins. These include the structural proteins core, E1 and E2 and the non-structural (NS) proteins p7, NS2, NS3, NS4A, NS4B, NS5A and NS5B. Together with the viral RNA the structural proteins form the viral particle, whereas the NS proteins are involved in viral morphogenesis [24,25]. HCV replication occurs in the cytoplasm in replicon complexes (RCs) at the so-called “membranous web” (MW), a characteristic hallmark of flavivirus-infected cells. These virus-induced compartments consist of lipid droplets (LDs) and rearranged ER (endoplasmatic reticulum)-derived membranes including single-, double-, and multi-membrane vesicles that allow a spatiotemporal separation of viral RNA translation, replication and assembly [26,27,28]. The double membrane vesicles (DMVs) are highly enriched in cholesterol and sphingolipids. In line with this, it has been described recently that HCV hijacks lipid transfer proteins to guarantee the establishment of cholesterol-enriched DMVs to maintain viral replication [29]. The assembly of the virions starts on the surface of core-associated cytosolic LDs (cLDs) in close proximity to the RCs [20]. Based on their heavy association with lipids and lipoproteins, the release of HCV virions has been linked to the lipoprotein pathway [25,30]. However, recent work indicated an exosome-dependent release via multivesicular bodies (MVBs), independent of the very-low-density lipoprotein (VLDL)-pathway [31,32].

## 3. Nuclear Factor Erythroid 2 (NF-E2)-Related Factor 2 (Nrf2)

Nrf2 was first described in 1994 by Moi et al. as a factor that binds to NF-E2 (nuclear factor erythroid 2) and AP-1 (activating protein 1) sites of the β-globin gene promoter [1]. Nrf2 belongs to the CNC (cap’n’collar) subfamily of bZIP (basic-region leucine zipper) transcription factors that bind to short cis-acting sequences, called ARE (antioxidant-response element) and EpRE (electrophile-response element), in the promoter regions of detoxifying genes encoding for phase I and phase II drug metabolizing enzymes, as well as phase III enzymes involved in cellular efflux [33,34,35,36,37,38]. Moreover, Nrf2 regulates the expression of proteins involved in the proteasomal pathway by binding to ARE sequences in the promoter region of the catalytic subunits PSMB5 and PSMB6 of the 20S proteasome [37,39,40]. Other members of the bZIP protein family are NF-E2 (nuclear factor erythoid 2) p45 [41,42], Nrf1 (NF-E2 related factor 1) [42,43], Nrf3 [42,44,45], Bach1 (BTB (Broad-complex, Tramtrack and Bric-à-brac) and CNC homology 1) [42,46] and Bach2 (BTB and CNC homology 2) [42,47].

### 3.1. Domain Structure of Nrf2

Human Nrf2 has a size of 605 amino acids and is composed of seven Neh (Nrf2-Ech homology) domains (Neh 1–7). Neh1 contains the CNC bZip domain that is essential for DNA binding and heterodimerization with sMaf (v-maf avian musculoaponeurotic fibrosarcoma) proteins (MafK, MafF, MafG) [48]. sMaf proteins are bZIP transcription factors that are classified as sMaf family based on their small size (160 aa, 18 kDa). They form heterodimers with CNC transcription factors (p45 NF-E2, Nrf1, Nrf2, Nrf3) and transcription factors of the Bach family (Bach1 and Bach2). As CNC and Bach proteins are not able to interact with DNA as homodimers, heterodimerization with sMaf proteins is required for their transcriptional regulation. sMaf proteins can also form homodimers that act as transcriptional repressors [49]. The Neh2 domain mediates Nrf2 degradation by binding of the repressor Keap1 (Kelch-like Ech-associated protein 1), an adaptor for the Cul3 (Cullin 3)-RBX1 (Ring-box-1)-dependent E3 ubiquitin ligase complex through its DLG and ETGE motifs [11,48,50]. Neh3, Neh4 and Neh5 act as transactivation domains through binding of CDH6 (chromo-ATPase/ helicase DNA-binding domain 6) [51], CBP (CREB (cAMP response element-binding protein)-binding protein) [52] and Rac3 (receptor-associated coactivator 3) [53]. Like Neh2, the Neh6 domain is involved in Nrf2 degradation by recruitment of the dimeric ubiquitin ligase βTrCP (β-transducin repeat-containing protein) through its DSGIS and DSAPGS motifs. This process is enhanced by phosphorylation of the DSGIS motif by GSK3β (glycogen synthase kinase 3β) [50,54]. The Neh7 domain is involved in repression of Nrf2 through binding to the RXRα (retinoid X receptor α) protein [55].

### 3.2. Nrf2 Regulation

Nrf2 is tightly controlled by a complex transcriptional/ epigenetic and (post-)translational network [7]. However, modulation of protein stability reflects the major regulatory mechanisms controlling Nrf2 activity [7]. Transcriptional activation occurs via autoregulation [56] and other transcription factors such as NF-κB (nuclear factor κB) [57], AhR (aryl hydrocarbon receptor) [58], PPARγ (peroxisome proliferator-activated receptor γ) [59], p53 [60], MEF2D (myocyte enhancer factor 2d) [61], c-Jun, c-Myc [62] and BRCA1 (breast cancer 1) [63], through binding to ARE- and XRE (xenobiotic response element)-like elements in its promoter region. In addition, miR (microRNA)-based mechanisms have been described as regulating Nrf2 activity through targeting Nrf2 mRNA and mRNAs that encode for proteins that regulate Nrf2 activity [64]. Epigenetic control of Nrf2 expression is mediated through hypermethylation of CpG sequences in its promoter region. In line with this, Nrf2 expression in prostate tumor of TRAMP (transgenic adenocarcinoma of mouse prostate) mice is repressed due to CpG methylation and H3 histone methylation [65].

Regarding translational control, under basal conditions, Nrf2 cap-dependent translation is suppressed whereas exposure to oxidative stress increases Nrf2 IRES-mediated expression [66]. Post-translational regulation can be divided into the classical canonical mechanism and the non-canonical mechanism [67,68].

#### 3.2.1. Canonical Nrf2 Activation

##### Keap1-Mediated Activation of Nrf2

Nrf2 regulation through the canonical mechanism is mainly mediated by Keap1. Keap1 belongs to the BTB (bric-a-brac) Kelch family that associates with the Cul3-RBX1-dependent E3 ubiquitin ligase complex [69]. Under basal conditions, dimeric Keap1 binds to the ETGE and the DLG motif within the N-terminal Neh2 domain of Nrf2. This complex constantly polyubiquitinates seven lysine residues within the Neh2 domain following proteasomal degradation via the 26S proteasomal pathway, to ensure low basal Nrf2 levels [48,50,67]. In addition, the ubiquitin-targeted ATP-dependent segregase p97 extracts ubiquitinated Nrf2 from the Cul3-RBX1 E3 complex for 26S proteasomal degradation [70]. Oxidative/ electrophilic stress triggers Nrf2 activation through oxidation of cysteine residues resulting in a conformational change of Keap1. The major cysteine residues involved in stress-sensing are Cys151, Cys273 and Cys288 [71]. It has been suggested that modifying Cys151 inhibits Keap1 under stress conditions whereas Cys273 and Cys288 regulate Keap1 under basal and stress conditions. In addition, further cysteine residues including Cys226, Cys434 and Cys613 are targeted by different stress inducers. However, the data are conflicting and need to be further investigated [69,72,73]. Keap1-cysteine oxidation results in an impaired Nrf2 ubiquitination, protecting Nrf2 from further proteasomal degradation. *De novo* synthesized Nrf2 accumulates and is translocated to the nucleus where it forms heterodimers with sMaf proteins that bind to ARE-sequences in the promoter region of its target genes. Nuclear translocation of Nrf2 through karyopherin α1 and karyopherin β1 is triggered by phosphorylation of Ser40 by PKC (protein kinase C) [74], Ser558 by AMPK (AMP-activated protein kinase) and AMPK-mediated inhibition of GSK3β [75] as well as other (undefined) kinases. To enhance binding with basic-region leucine zipper proteins, acetylation by CBP acetylase [76] and p300/CBP [77] and SUMOylation by UBC9 (E2 SUMO (small ubiquitin-like modifier)-conjugating enzyme) [78] of nuclear Nrf2 occurs. Binding of the co-activators CDH6 [51] and Rac3 [53] further enhances Nrf2 activity.

##### βTrCP-Mediated Activation of Nrf2

Alternatively, Nrf2 activity is controlled by the dimeric ubiquitin ligase βTrCP that acts as an adaptor for the Skp1 (S-phase kinase-associated protein 1)-Cul1-Rbx1 E3 ubiquitin ligase complex. Phosphorylation of a group of serine residues within the Neh6 domain by the constitutive active serine/threonine kinase GSK3β results in the formation of a phosphodegron, which tethers the Skp1-Cul1-Rbx1 E3 ligase complex resulting in Nrf2 ubiquitination and subsequent proteasomal degradation. βTrCP association is mediated through the DSGIS and DSAPGS motifs within Neh6 of Nrf2. Additional phosphorylation of the DSGIS motif (Ser344, Ser347) by GSK3β thereby promotes inhibition of Nrf2 activity. Inhibitory phosphorylation of GSK3β occurs via the PKB (phosphatidylinositol 3-kinase (PI3K) kinase B)/AKT (protein kinase B) pathway, thus activation of the PI3K or PKB/AKT mediates Nrf2 activation [54,67,79,80]. In addition, GSK3β indirectly triggers nuclear export of Nrf2 followed by subsequent ubiquitination and proteasomal degradation through targeting the subcellular localization of the tyrosine kinase Fyn that catalyzes Nrf2-phosphorylation (Tyr576) [81].

##### Hrd1-Mediated Activation of Nrf2

Another ligase involved in Nrf2 regulation is the E3 ubiquitin ligase Hrd1 that is part of the IRE1 (inositol-required protein 1) pathway of the UPR (unfolded protein response). Upon induction of ER-stress, IRE1 catalyzes the splicing of XBP1 (X box-binding protein 1) to form the active transcription factor sXBP1 (spliced XBP1) and transcriptional upregulation of Hrd1. It has been described recently that patients with chronic liver cirrhosis fail to inactivate high levels of ROS (reactive oxygen species) due to impaired Nrf2/ARE-signaling. In cirrhotic livers, the XBP1-Hrd1 pathway is upregulated resulting in increased Nrf2 ubiquitination and proteasomal degradation, hence Nrf2-mediated protection is inhibited [82].

#### 3.2.2. Non-Canonical Nrf2 Activation

Recently, the non-canonical pathway of Nrf2 activation has been described. Nrf2 activation in this pathway is mediated by proteins that compete with Nrf2 for Keap1 binding, thus stabilizing Nrf2. These proteins harbor a motif similar to the ETGE motif in the Neh2 domain of Nrf2. Nrf2-binding proteins known so far are p21 and BRCA1 (breast cancer type 1 susceptibility protein); Keap1-binding proteins are p62/SQSTM1 (sequestosome 1), DPP3 (dipeptidyl peptidase III), WTX (Wilms tumor gene in chromosome X), ProTα (Prothymosin α) and PALB2 (partner and localizer for BRCA2) (for a detailed review of Nrf2- and Keap1-binding proteins see [67,68]). The best studied protein involved in the non-canonical Nrf2 regulation is the Keap1-binding protein p62/SQSTM1 (hereafter referred to as p62). p62 is a stress-inducible multi-domain protein that acts as a signaling hub for a variety of cellular processes. p62 is ubiquitiously expressed in various cell types and can be found in the cytoplasm of the cell as well as in the nucleus, autophagosomes and lysosomes. During selective autophagy, p62 serves as cargo receptor for autophagic degradation and ubiquitinated cargo through interaction with its LIR (LC3 interacting region) and UBA (ubiquitin-associated) domains [83,84]. Binding of ubiquitinated proteins and damaged mitochondria leads to formation of p62 inclusion bodies that are degraded via autophagy [85,86,87]. Besides, p62 competes with Nrf2 for binding to Keap1 through its KIR (Keap1-interacting region) domain. Phosphorylation of Ser349 by mTORC1 (mammalian target of rapamycin complex 1), TAK1 (transforming growth factor β-activated kinase 1) and other undefined kinases enhances binding of p62 to Keap1 [88,89]. p62-Keap1-association sequesters Keap1 in p62 inclusion bodies, guiding them towards autophagosomal degradation. Hence, Nrf2 is withdrawn from proteasomal degradation resulting in nuclear translocation and expression of its target genes [81]. Other kinases involved in p62-mediated autophagosomal clearance are CK2 (casein kinase 2) (Ser403) [90], TBK-1 (TANK-binding kinase 1) (Ser403) [91], Sestrin2-ULK1 (Unc-51-like kinase 1) complex (Ser403) [92] and ULK1 (Ser407) [93], that increase the affinity of p62 to ubiquitin. In addition, TRIM16 (tripartite motif containing 16) has been described as being essential for phosphorylation of p62 on Ser349. TRIM16 is involved in protein turnover as it interferes with components of the autophagosomal pathway involved in autophagy initiation (ULK1), phagophore elongation (ATG16L1) and LC3 [94].

Upon increased levels of ROS, Nrf2 activates p62 expression as it harbors an ARE sequence in its promoter region. Otherwise, p62 mediates Nrf2 expression, thereby creating a positive feedback loop [95]. However, p62-dependent activation of Nrf2 induces expression of cytoprotective genes, thus protecting the cell from oxidative stress [96]. In addition, Keap1 competes with LC3 for binding to p62, and thereby inhibits autophagic degradation of p62 [81]. Hence, increased p62 levels indicate an impaired autophagic flux [97,98]. In line with this, Nrf2 activation due to a defect in the autophagic pathway following p62 accumulation has been associated with cancer and drug resistance [13,14,88,97,99].

Summing up, the p62-mediated non-canonical activation of Nrf2 is tightly connected to the autophagosomal pathway based on a direct interaction of p62 and Keap1 [88,95,97,98]. Interaction of p62 with ubiquitinated proteins mediates the formation of protein aggregates that are finally degraded via the autophagosomal pathway. In line with this, p62 triggers autophagosomal turnover of Keap1 following activation of Nrf2. Based on the tight connection of these two mechanisms, the autophagic pathway will be described in more detail in the following.

## 4. Autophagy

Autophagy (“self-eating”) is a highly regulated catabolic process to maintain cellular homeostasis by degrading intracellular components such as long-lived proteins, protein aggregates, damaged organelles and pathogens as a response to different stress signals, such as nutrient deprivation, viral infection and ROS [100]. Formation of the autophagosome can be divided into three steps including nucleation, expansion and closure [101], starting with the formation of an IM (isolation membrane) (originally termed the phagophore) at PAS (phagophore assembly sites) which increase to enclosed double-membrane vesicles known as autophagosomes. Autophagosomal membranes originate from different sources including recycling endosomes, Golgi Apparatus, plasma membrane, mitochondria and ER [102,103,104]. The autophagosomes directly fuse with lysosomes to form autophagolysosomes where the cargo is digested by lysosomal hydrolases and the acidic environment [105]. Alternatively, autophagosomes fuse with MVBs (multivesicular bodies) to form an amphisome that finally can fuse with a lysosome [106]. The degraded cargo is released into the cytoplasm where it serves as energy source or de novo synthesis of molecules [107]. Autophagic turnover is tightly regulated by a set of >30 autophagy-regulated genes (Atg) [102,108]. Autophagy initiation is controlled by the mammalian target of rapamycin complex 1 (mTORC1). Under nutrient-rich conditions, mTORC1 phosphorylates the ULK1/2-complex (Unc-51-like kinase 1 and 2), hence, inhibiting autophagy. During nutrient deprivation, autophagy is induced due to inhibition of mTORC1 by AMPK (AMP-activated protein kinase) following mTORC1 release from the ULK1/2-complex and subsequent activation of ULK1 and ULK2 kinases. The activated ULK1/2-complex is shuttled to the phagophore nucleation site where it activates the PI3K-complex (composed of PI3K, Vps34 (vacuolar protein sorting 34), p150, ATG14L and BCLN1 (Beclin1), catalyzing an PI3P (phosphatidylinositol-3-phosphate)-enriched environment [103,109]. PI3P further recruits DFCP1- (double FYVE-containing protein 1) and WIPI- (WD-repeat domain phosphoinositide-interacting) proteins triggering phagophore nucleation. In addition, mammalian Atg9 (mAtg9) initiates binding of DFCP1 to autophagosomes. mAtg9, the only multi-spanning transmembrane protein of the core autophagic machinery, cycles between the PAS and multiple organelles and is essential for expansion of the IM [110,111,112,113]. Two ubiquitin-like conjugation systems Atg5-Atg12-Atg16L and Atg4-Atg3-LC3II/GABARAP (gamma-aminobutyric acid receptor-associated protein participate) catalyze the expansion of the IM to finally create an enclosed autophagosome. At first, Atg12 is conjugated to Atg5 by an ubiquitin-like reaction through Atg7 (E1-like) and Atg10 (E2-like). The Atg5-Atg12 conjugate then binds to Atg16L to generate the membrane-associated Atg12-5-16L complex (E3-like). In a next step, LC3 (microtubule-associated protein 1 light chain 3) and GABARAP are cleaved by Atg4 at the C-terminus. Finally, the cleaved cytosolic LC3-I/GABARAP-I is activated by Atg7 (E1-like), transferred to Atg3 (E2-like) and conjugated to PE (phosphatidylethanolamine) to gain the lipidated LC3-II/GABARAP-II that is localized to autophagosomal membranes. The lipidated LC3/GABARAP remains membrane-associated until it gets degraded in the autophagolysosome or is removed by Atg4-mediated cleavage [114,115,116]. The maturation of the enclosed autophagosomes further involves the release of Atg proteins from the autophagosomal membranes, triggered by PI3P turnover [117] and fusion of the autophagosome with lysosomes to form an autophagolysosome. Autophagosomes may also fuse with MVBs to form an amphisome that finally fuse with a lysosome (for a detailed review of autophagy and components/ molecules involved in membrane trafficking of autophagosomal membranes see [101,104]).

## 5. Reactive Oxygen Species (ROS) in HCV-Infected Cells

HCV is associated with oxidative stress in liver cells that results in increased levels of ROS that encompass superoxide anions (O^2−●^), hydroxyl radical (OH^●^) and hydrogen peroxide (H_2_O_2_). To date, the viral proteins core, E1, E2, NS3, NS4B and NS5A, has been described to interfere with pathways and enzymes that trigger production of ROS [118,119]. These include mitochondrial dysfunction due to Ca^2+^-redistribution, activation of NADPH (nicotinamide adenine dinucleotide phosphate ) oxidases (NOX1 and 4), enhanced CYP2E1 (cytochrome P450 2E1) and Ero1α (ER oxidoreductin 1α) expression and induction of ER stress and the UPR (Figure 1.). The HCV core and NS5A proteins are considered to be main activators of ROS production [118,120]. They can be found on the surface of LDs, the nucleus, the ER, mitochondria and MAMs (mitochondrial-associated membranes) [121,122,123,124]. ROS-mediated mitochondrial dysfunction in HCV-infected cells is mainly caused by core, NS5A and marginal by other viral proteins [125,126,127,128]. Although the ER presents the main intracellular Ca^2+^-storage, mitochondria as well have a high Ca^2+^-storage capacity [129]. In line with this, mitochondria participate in Ca^2+^-signaling and intracellular Ca^2+^-homeostasis. Increased mitochondrial Ca^2+^ concentrations are associated with increased electron transport, production of ROS, and opening of the mPTP (mitochondrial permeability transition pore) [130]. Binding of the HCV core protein to the OMM (outer mitochondrial membrane) with its hydrophobic C-terminus triggers mitochondrial Ca^2+^-influx via the MCU (mitochondrial Ca^2+^-uniporter) that is located in the IMM (inner mitochondrial membrane). This results in an increased MPT (mitochondrial permeability transition), inhibition of complex I of the respiratory chain and release of cytochrome c which in turn activates apoptosis [131,132,133,134]. Mitochondria are tightly connected to the ER via MAMs that facilitate Ca^2+^-fluxes and the transfer of membrane bound lipids [135]. It has been described that expression of the core protein triggers ER Ca^2+^-efflux through induction of ER-stress and inhibition of SERCA (sarcoplasmic/endoplasmic reticulum calcium ATPase 2) [136]. Likewise, NS5A induces ER Ca^2+^-efflux [137,138]. Moreover, interaction of NS5A with the ER-localized lipid kinase PI4K4 (phosphatidylinositol 4-kinase IIIα) tethers the ER with mitochondria and triggers mitochondria fragmentation [127,139,140]. Ca^2+^-fluxes are further controlled by the oxidase Ero1α that is enriched in MAMs. Ero1 proteins, together with the PDI (protein disulfide isomerase), are ER-resident proteins involved in protein folding as they catalyze the formation of disulfide bonds, thereby generating H_2_O_2_ as a by-product. Expression of these proteins is increased after activation of the UPR [141,142]. Recently the core protein was found to increase Ero1α expression following enhanced ER Ca^2+^-efflux, mitochondria Ca^2+^-influx and formation of superoxide anions [143]. In addition, the viral proteins E1 and E2 [144] and NS4B [145,146] have been reported to induce ER-stress and the UPR and possibly interfere with Ca^2+^-homeostasis [119]. Formation of ROS in HCV-infected can be further triggered by increased expression of NADPH oxidases (NOX). The NOX family of NADPH oxidase is a multisubunit transmembrane enzyme complex that catalyzes the formation of ROS in form of superoxide anions (O^2−●^) and hydrogen peroxide (H_2_O_2_). The superoxide anions are generated by the transfer of electrons from NAD(P)H to O_2_, hydrogen peroxide through dismutation of the superoxide [147]. In HCV-infected cells the NS5A and core protein induce NOX1 and NOX4 expression resulting in elevated ROS formation. This is mediated by HCV-induced nuclear localization of NOX4 and accumulation of TGF-β1 (transforming growth factor beta-1) in HCV-infected cells [143,148,149,150]. Furthermore, NS3 has been reported to induce the expression of NOX in human monocytes [128]. Another mechanism in HCV-induced ROS formation is the enhanced expression of CYP2E1. CYP2E1 is highly expressed in the liver, where HCV morphogenesis takes place. The enzyme is localized on the ER and the Golgi and is involved in the metabolism of drugs, hormones and xenobiotics [151]. In addition, CYP2E1 is a major component of the MEOS (microsomal ethanol oxidizing enzymes) that catalyzes the conversion of ethanol to acetaldehyde resulting in production of superoxide anions (O^2−●^) and hydrogen peroxide (H_2_O_2_) [152]. In HCV-infected cells, core and NS5A have been described to induce expression of CYP2E1 thereby contributing to elevated levels of oxidative stress [143,150]. Moreover, it has been described by Wen et al., that in stably core- and CYP2E1-overproducing HepG2 cells increased ROS formation and sensitization to cell injury due to GSH (glutathione) depletion can be detected [153].

## 6. Interference of HCV with the Nrf2/Keap1-Signaling Pathway

Cells have evolved efficient strategies to counteract ER- and oxidative-stress. An imbalance in protein-homeostasis at the ER triggers the induction of the UPR to increase expression of genes encoding for proteins involved in ERQC (ER protein quality control), and to reduce the influx of proteins to ensure proper protein folding. The UPR signaling network includes three stress sensors: the IRE1, PERK (protein kinase (PKR)-like ER kinase), and ATF6 (activating transcription factor 6) (for a detailed review of the UPR see [154,155]). ER-stress further triggers activation of autophagy [100,156]. In addition, activation of the UPR due to elevated oxidative stress levels results in a direct PERK-dependent Nrf2-activation [157] or indirect Nrf2-activation via the IRE1α/JNK (c-Jun-N-terminal kinase) pathway [158]. Infection with HCV is accompanied by a massive rearrangement of ER-derived membranes and dysfunctional ER protein-homeostasis of the host cell. Based on this, it is described that HCV infection leads to induction of ER-stress and the UPR in vitro and in vivo (for a detailed review see [159,160,161]). Besides, HCV-infection interferes with the autophagosomal pathway. Activation of autophagy in HCV-infected cells occurs either by direct interaction of HCV proteins with components of the autophagy machinery or indirectly through induction of ER- and oxidative stress (for a review see [162,163]).

Another signaling-pathway involved in maintaining cellular homeostasis in response to oxidative/electrophile stress is the Nrf2/Keap1-signaling pathway [7,68,164,165]. Infection with HCV is associated with oxidative stress in liver cells that results in increased levels of ROS [119]. In this regard, HCV has been described as interfering with the Nrf2/Keap1-signaling pathway. However, the data regarding Nrf2-modulation are conflicting. Interestingly, Carvajal-Yepes et al. identified a defect in this pathway in HCV-infected cells, based on a core-mediated delocalization of sMaf-proteins from the nucleus to the ER-associated RCs, where they are trapped by binding to NS3. As a consequence, the delocalized sMaf-proteins bind to Nrf2 and prevent the translocation of Nrf2 into the nucleus resulting in impaired expression of cytoprotective genes [166]. The increased ROS levels further trigger the phosphorylation of p62 on Ser349 (pS(349) p62) [88]. Due to the defect in the Nrf2/Keap1-signaling pathway, the pS(349) p62-mediated activation of Nrf2 cannot compensate the increased ROS levels resulting in activation of autophagy that favors the release of HCV particles [167] (see Figure 1). In addition, NS5A interferes with Nrf2-activation via a crosstalk with the MAPK (mitogen activated protein kinase) signaling cascade. The MAPK/ERK (extracellular signal-regulated kinase) signaling pathway triggers phosphorylation of Nrf2 that leads to Keap1-Nrf2 dissociation and subsequent Nrf2-activation [37]. In this regard, it has been described that NS5A recruits cRaf to the ER-associated RCs resulting in activation of cRaf and NS5A phosphorylation that is essential for efficient HCV replication. However, despite cRaf activation, no activation of the MAPK signaling cascade could be observed resulting in impaired Nrf2-activation [168,169,170] (see Figure 1). In line with this, transcriptome analysis HCV-infected cells revealed a decreased expression of the Nrf2-regulated genes NQO1 (NAD(P)H: quinone oxidoreductase 1), ephx1 (epoxide hydrolase 1), cat (catalase), GCLC (glutamate-cysteine ligase catalytic subunit) and other enzymes of the glutathione metabolism [171,172]. Furthermore, in liver-biopsy samples of HCV-infected patients and stably core-producing HepG2 cells, a decreased expression of the Nrf2-regulated HO-1 could be detected [173].

Conversely, Burdette et al. revealed an HCV-mediated Nrf2-activation due to an impaired ER-Ca^2+^-homeostasis, increased ROS levels and Nrf2-phosphorylation through activated MAP kinases [174]. A study by Jiang et al. described an activation of Nrf2 in HCV-infected Huh7.5.1 cells and liver biopsy samples from CHC (chronic hepatitis C) patients based on the inhibitory GSK3β phosphorylation [175]. In addition, Ivanov et al. identified the HCV-proteins core, E1, E2, NS4B and NS5A to mediate Nrf2-activation with core and NS5A being the key regulators. Expression of NS5A and the core protein resulted in transcriptional and translational upregulation of HO-1 and NQO1 via two mechanisms [120]: PKC catalyzes phosphorylation of Nrf2 upon elevated oxidative stress levels, whereas CK2 and PI3K trigger Nrf2-phosphorylation by ROS-independent mechanisms [118]. In addition, an activation of HO-1 and NQO1 in HCV-replicating cells and patients with chronic liver diseases has been described [176,177,178,179]. The conflicting results in literature may be ascribed to different HCV models and experimental setups. In this context, the extent of oxidative-stress differs during the acute and chronic HCV-infection and thereby contributes to a different induction of the cellular stress response. In line with this, Anticoli et al. observed high ROS levels during the acute phase of infection that are accompanied with high rates of viral replication and transcriptional NQO1-activation. In contrast, the chronic phase of HCV-infection is characterized by reduced ROS production to favor the establishment of a chronic infection. Thereby, induction of the Nrf2/Keap1-signaling pathway is a crucial requisite to overcome the HCV-induced oxidative stress as the establishment of a chronic infection requires the survival of the hepatocytes [160,174,179].

Thus, based on the non-canonical p62-mediated activation of Nrf2, the Nrf2/Keap1-signaling is linked to the autophagosomal pathway [88,95,97,98]. A defect in the autophagic pathway in hepatocytes, mediated by deletion of Atg5 and Atg7, resulted in accumulation of p62 in ubiquitin-containing inclusion bodies accompanied by increased levels of oxidative stress, DNA damage, liver inflammation, fibrosis and the development of liver tumors [180]. In line with this, liver-specific Atg7 knockout (KO) mice exhibit a massive accumulation of p62 and subsequent Nrf2-activation [181]. However, simultaneous deletion of p62 and Atg7 suppressed the size of liver tumors and deletion of Nrf2 in Atg5 KO mice restored the above described pathological phenotype [180,182].

The expression of p62 is increased in many human cancers and chronic liver diseases [84]. In 40%–50% of patients suffering from HCC (hepatocellular carcinoma) an increase of mTORC1 activity has been detected, resulting in enhanced p62 phosphorylation and enhanced Nrf2 activation. However, this phenotype is associated with poor prognosis [183,184]. Furthermore, an association between a dysfunctional autophagy and Nrf2 activation in HCC has been described [99]. In this regard, persistent activation of Nrf2 is associated with accumulation of p62 and development of HCC [185].

## 7. Impact of ROS in HCV-Associated Pathogenesis

Chronic infection with HCV is a leading cause of the development of hepatocellular carcinoma (HCC) [186]. However, the development of an HCV-associated HCC arises after decades and requires persistent inflammation of the liver that is accompanied by chronic cycles of hepatocytic cell death and liver regeneration that finally lead to liver damage and loss of liver function [186,187]. Remarkably, the liver and the skin are the only organs of the body that have the ability to fully regenerate (“The myth of Prometheus”). After liver injury the loss of functional liver tissue is compensated by proliferating hepatocytes to reconstitute the original mass. Liver regeneration can be divided into three phases: activation, proliferation, and termination. Upon liver damage, 95% of the quiescent hepatic cells re-enter the cell-cycle from the G0, through the G1, to the S phase. This is triggered by the release of pro-inflammatory cytokines, such as TNF-α (tumor necrosis factor α) and IL-6 (interleukin 6) and the subsequent activation of the transcription factors NFκB, AP-1 and STAT3 (signal transducer and activator of transcription 3) [188]. Proliferation of the hepatocytes is further stimulated by growth factors such as HGF (hepatocyte growth factor) [189], ligands of the EGFR- (epidermal growth factor receptor), with EGF (epidermal growth factor) and TGF-α being the most prominent [190,191], and FGF (fibroblast growth factor) [192]. In addition, IGF-I (insulin-like growth factor I) has been described to participate in liver regeneration [193]. After reconstitution of the liver mass, TGF-β and activin inhibit hepatocyte proliferation returning the liver into a quiescent state [194].

In addition, hepatic fibrosis as a wound-healing process is involved in liver repair that is characterized by deposition of ECM (extracellular matrix) (e.g., collagen, proteoglycans) by activated HSC (hepatic stellate cells) and myofibroblasts [195]. Activation of HSCs is promoted by cytokines such as TGF-β1, PDGF (platelet-derived growth factor) and CTGF (connective tissue growth factor) [196]. Persistent HCV-infection triggers an excessive production of ECM leading to liver fibrosis that, over decades, results in replacement of functional hepatocytes by non-functional scar tissue and the establishment of liver cirrhosis and HCC [187]. In light of this, it has been described that the HCV core, NS3/4A, NS4B, NS5A induce production of TGF-β1 through increased ROS-levels and interference with the mitochondrial Ca^2+^-homeostasis which leads to progression of liver fibrosis [197]. Wu et al. reported that free core-protein triggers activation of HSCs via an ObR (obese receptor)-dependent JAK2 (Janus kinase 2)/STAT3, AMPKα (AMP-activated protein kinase), and AKT-signaling pathway [198]. In addition, the HCV E2-protein triggers proliferation of HSCs through a JAK-dependent upregulation of collagenα-I and oxidative stress [199]. Increased oxidative stress in hepatocytes can be further induced via CYP2E1 [200]. HCV core and NS5A induce CYP2E1-mediated oxidative stress that in turn triggers HSC proliferation [143,150]. However, the main factor that triggers the onset of HCV-associated pathogenesis is the induction of oxidative stress. Increased ROS-levels result in DNA damage and accumulation of DNA damage may lead to genetic mutations and mutagenesis [201]. In line with this, the Nrf2/Keap1-signaling pathway plays an essential role in maintaining cellular redox-homeostasis. However, dysregulation of the Nrf2/Keap1-signaling pathway is associated with the progression of cancer [202,203]. Over time, a dual role of Nrf2 in carcinogenesis has evolved. During the early stage of carcinogenesis, Nrf2 functions as a tumor suppressor as it eliminates increased ROS levels and stimulates GSH synthesis to promote cell survival under physiological conditions [204]. In the tumor microenvironment, activation of Nrf2 is promoted by the tumor suppressor genes BRCA1 and protein p21 [205,206] and is blocked by Fyn-mediated degradation [207]. The cyclin-dependent kinase inhibitor p21/waf1 plays an essential role in controlling the cell-cycle, DNA repair, cell differentiation, apoptosis and senescence. Under physiological conditions, induction of p21/waf1 results in cell-cycle arrest [208,209,210]. The p21-mediated Nrf2-activation is weakened by HCV core- and NS5A-proteins that interact with the p21/waf1 protein and thus downregulate its expression resulting in cell proliferation [211,212,213].

However, persistent and constitutive Nrf2-activation has been associated with the progression of liver cancer [83,214]. Persistent activation of Nrf2 is promoted by at least five different mechanisms: (1) impaired Nrf2/Keap1-interaction due to somatic mutations in Nrf2, Keap1 or Cul3 or Nrf2 exon skipping, (2) increased Nrf2-transcription, (3) reduced Keap1-levels, and (4) stabilization of Nrf2 by proteins that compete with Nrf2 for Keap1 binding e.g., p62 or (5) Keap1 succination (for a detailed review see [204]).

Besides its role in the cellular stress response, Nrf2 plays an essential role in tissue repair [10]. In Nrf2 knockout mice, liver regeneration was significantly delayed after partial hepatectomy. This effect is based on a ROS-mediated resistance of the insulin/IGF-1 (insulin growth factor-1) receptor signaling. The ability of insulin to stimulate glucose uptake is essential in controlling glucose homeostasis [215]. In healthy individuals, uptake of glucose into muscle and adipose tissues induces the secretion of insulin from β-cells of the pancreatic islets of Langerhans. A simplified model of the basal system acts as follows: insulin binds to its receptor followed by subsequent phosphorylation and activation of the IRS-1/ IRS-2, activation of the (PI3K)/AKT pathway and translocation of GLUT4 (glucose transporter 4) to the PM (plasma membrane). HCV interferes with different steps of this pathway. Increased ROS-levels activate serine/threonine kinases that catalyze phosphorylation of both IRS-1 and IRS-2 (insulin receptor substrate) resulting in insulin/IGF resistance. Ser/Thr-phosphorylated IRS dissociates from the insulin receptor, thereby preventing the activation of IRS by Tyr-phosphorylation and leads to an impaired (PI3K)/ AKT signaling [216]. Further downstream in this pathway, JNK has been proposed as one of the inhibitory Ser/Thr kinases, as it is known to respond to enhanced ROS-levels [217,218]. Due to the impaired Nrf2/ Keap1-signaling during HCV-infection the increased ROS-levels may contribute to the development of insulin resistance through JNK-mediated Ser/ Thr-phosphorylation of IRS-1 and IRS-2 (Figure 1.).

Accordingly, chronic infection with HCV is associated with insulin resistance and the progression of T2DM (type 2 diabetes mellitus), hepatic steatosis and liver fibrosis and resistance to antiviral treatments [219]. Alongside ROS-mediated insulin resistance, several other mechanisms have been identified as a HCV-induced cause for this pathogenic effect. In HCV core tg (transgenic) mice, higher levels of plasma glucose and resistance to insulin could be detected. This effect was deduced to increased TNFα-levels, which inhibit Tyr-phosphorylation of IRS-1 and are consistent with previous results of patients suffering from chronic CHC [220]. In addition, expression of the core protein increases SOCS3 (suppressor of cytokine signaling 3) and thereby promotes proteasomal degradation of IRS-1 and IRS-2 in core tg mice and transiently core-transfected human hepatocytes [221]. It has been described recently that HCV core activates JNK- and MAPK-pathways. In line with this, core induces phosphorylation of IRS-1 on Ser^312^ (JNK-mediated) accompanied by decreased glucose uptake and degradation of IRS-1 [222]. The HCV-dependent impairment of the (PI3K)/AKT-signaling pathway further modulates translocation and activation of FoxO1 and FoxA2 (Forkhead box transcriptional regulators) and regulates their metabolic functions. FoxO1 regulates the expression of genes involved in glucose and lipid metabolism. Insulin-mediated AKT-dependent phosphorylation of FoxO1 interferes with its DNA-binding. In contrast, insulin inhibits phosphorylation of FoxA2 that mediates lipid metabolism during fasting [223]. Moreover, infection with HCV has been proposed to suppress expression of TSC-1/ TSC-2 (tuberous sclerosis complex) and activate the mTOR/S6K1 pathway resulting in IRS-1 degradation through Ser^1102^ phosphorylation. HCV-dependent effects on glucose-metabolism is further established via interference with GLUT4 and PCK2 (phosphoenolpyruvate carboxykinase 2) [224]. In line with this, expression of PGC1α (peroxisome proliferator-activated receptor-gamma co-activator 1α), a key transcription factor in gluconeogenesis, was found to be strongly induced in HCV-infected cells due to increased oxidative stress in HCV-infected cells [225,226,227].

Furthermore, interference of HCV with insulin signaling seems to be genotype-specific as HCV core of genotype 3a induced IRS-1 degradation via downregulation of PPARγ and upregulation of SOCS-7, whereas the HCV core of genotype 1b activates the mTOR-pathway [228,229].

In addition, p62 has been identified to interfere with liver regeneration. Liver regeneration in liver-specific Atg5 KO mice was significantly delayed after partial hepatectomy (PH) [230]. KO of Atg5 was associated with massive accumulation of p62 and subsequent Nrf2-activation [182]. In line with this, constitutive Nrf2 activation impaired hepatocyte proliferation after PH and increased apoptosis due to an increase in the cyclin-dependent kinase inhibitor p15 and the pro-apoptotic protein Bcl2l 1 (Bim) [231]. In contrast, PH in mouse steatotic livers resulted in decreased expression of p62 and less Nrf2-activation. Consequently, the elevated ROS levels further trigger liver damage and impair liver regeneration [232].

## 8. The Hepatitis B Virus (HBV)

The human hepatitis B virus (HBV) belongs to the family of Hepadnaviridae. HBV infects with high tissue and species specificity human hepatocytes. The HBV virion is a spherical particle, 42 nm in diameter, HBV is an enveloped virus. The envelope is composed by host cell derived lipids and HBV surface antigen (HBsAg), which encompasses three different surface proteins: the large HBV surface protein (LHBs), the middle surface protein (MHBs) and the small surface protein (SHBs) [233,234]. The envelope surrounds the icosahedral nucleocapsid that is formed by the core protein. In the intact virion the nucleocapsid harbours the partially double-stranded circular DNA genome with a size of about 3.2 kB. The HBV genome encodes at least for four open reading frames: the polymerase, the surface proteins (LHBs, MHBs and SHBs), the core protein, including its secretory variant HBeAg, and the regulatory protein HBx.

In addition to virions subviral particles, exclusively assembled by HBsAg in the shape of spheres (also designated as 22 nm particles) and filaments lacking any capsid and viral DNA, are released by the infected cell. While spheres are almost exclusively formed by SHBs the filaments are characterized by a larger amount of LHBs [235,236]. The length of filaments varies between 50 and 200 nm. While spheres are secreted by the general secretory pathway, the release of virions and filaments depends on the ESCRT (endosomal sorting complex required for transport)-system and occurs via MVBs [237,238,239].

Although a prophylactic vaccine was developed in the early 1980s, there are at present two billion people who underwent an acute infection by HBV and about 240 million patients suffering from chronic infection with HBV worldwide [233,240]. Chronic HBV infection can cause liver fibrosis and ultimately cirrhosis [241,242]. Chronic hepatitis B virus infection is the leading cause for the development of human hepatocellular carcinoma (HCC). There are more than 700,000 deaths annually associated with chronic HBV infection. In many cases chronic HBV infection is characterized by a weak and inefficient cellular immune response which fails to clear completely HBV from the liver [243,244]. Thus, a circle between inefficient elimination of infected cells, regeneration of hepatocytes reinfection and again inefficient elimination/control of the infection starts that later gets out of control. Functional liver tissue is replaced by fibroblast leading to the excessive formation of connective tissue and subsequent fibrosis, cirrhosis and HCC [245].

## 9. Generation of ROS in HBV-Replicating Cells

The mechanisms leading to the generation of ROS in HBV replicating cells are not fully understood. HBx, HBsAg and HbcAg are described as viral proteins that are involved in ROS formation [119].

HBx is considered as an important factor triggering the formation of ROS. This is supported by the observation that HBx, in addition to its localization in the nucleus and in the cytoplasm, was found to be associated with mitochondria [246,247]. Several domains of HBx, aa 68–117 [248], aa 111–117 [249] and aa 121–154 [250] were described to mediate the interaction of HBx with the outer mitochondrial membrane. In this context, it is reported that the association of HBx with the outer mitochondrial membrane leads to membrane permeabilization [251]; the interaction of HBx with cardiolipin is described as causative factor for the loss membrane integrity [252]. Membrane permeabilization causes a breakdown of the mitochondrial membrane potential and thereby can lead to enhanced ROS production. Moreover, it was reported that HBx affects the activity of respiratory complexes I, II, IV and V by decreasing the expression of their subunits [253]. This was reported to be associated with a loss of the mitochondrial membrane potential. Apart from the effect on the expression of subunits of the respiratory chain, HBx was described to directly bind to COXIII (cytochrome c oxidase) that is part of the cytochrome c oxidase respiratory complex IV based on yeast-two-hybrid experiments [247,254]. However, it has to be considered that HBx seems to be localized on the outer mitochondrial membrane while COXIII is localized at the inner membrane.

In addition, voltage-dependent anion channel 3 (VDAC3) was described as binding partner of HBx [255]. VDAC3 is localized at the outer mitochondrial membrane and is involved in the regulation of the PTP [256,257]. Dysregulation of PTPs is a well established mechanism for ROS induction. In addition, there are reports that HBx triggers a significant increase of Ca^2+^ levels in the cytoplasm and in the mitochondria that might dysregulate PTP function in addition [258,259].

Overexpression of LHBs was found to be associated with its retention in the ER and intracellular accumulation [260,261]. Due to the strong ER overload, the formation of so called ground glass hepatocytes occurs. ER overload is associated with ER-stress and induction of the UPR [262]. The UPR leads to the release of Ca^2+^ into the cytoplasm that triggers the induction of ROS. In addition, there were reports that C-terminally truncated forms of the middle hepatitis B virus surface protein (MHBst) that are retained at the ER, trigger ROS production due to ER overload [262,263,264]. This was considered as a causative factor for the subsequent activation of NF-κB [263,265]. Further analyses however revealed that the ER retention seems to be less relevant for the NF-kB activation by these C-terminally truncated variants [266,267,268,269]. Moreover, analysis of JNK2 activity in MHBst producing cells or transgenic mice provided no evidence for the induction of cell stress due to ER overload or elevated ROS levels as investigated by oxyblots or determination of the 8-OHdG-level [268,269]. Detailed analyses identified the PreS2 domain as the minimal domain that is causative for the function as regulatory protein. As the PreS2 domain is not associated with the ER membrane and represents a cytoplasmic protein, this argues against ER overload and subsequent stress response leading to an activation of NF-kB. The membrane topology is crucial for the functionality of the PreS2 domain to serve as regulatory protein [267,268,269]. If the PreS2 domain faces the lumen of the ER, as in case of MHBs, N-glycosylation at Asn4 in the PreS2 domain occurs and no transcriptional activator function is exerted [267,268]. For LHBs a dual membrane topology was described due to a posttranslational translocation of the PreS1PreS2 domain [270,271,272]. Moreover, these C-terminal-truncated MHBs proteins (MHBst) that exert a transcriptional activator function, display a cytoplasmic orientation of the PreS2 domain [267,268]. In accordance to this, overproduction of the PreS2 domain that lacks any membrane association, was found to be sufficient for the function as regulatory protein [267]. In the cytoplasm, the PreS2 domain interacts with the classic PKC isoforms a and b and thereby induces the activation of PKC that is transduced via the c-Raf-MEK-Erk signal transduction cascade [269,273].

## 10. Interference of HBV with the Nrf2/Antioxidant Response Element (ARE) System

Recent reports revealed that a loss of Nrf2 is associated with an impaired liver regeneration [9,274,275,276,277]. The lack of Nrf2 prevents the expression of a variety of cytoprotective genes that in part are involved in the detoxification of reactive oxygen species [37,278].

Further analyses provided evidence that the impaired liver regeneration in Nrf2-deficient mice is associated with an elevated ROS level. The elevated ROS level trigger an activation of JNK that triggers a phosphorylation of IRS-1/-2 at the Ser/Thr-residues 303. Dependent on its phosphorylation, IRS-1/-2 acts as a switch mediating insulin receptor-dependent signal transduction pathways. Tyrosine phosphorylation of IRS-1/-2 is involved in the transduction of proliferative signals induced by activated insulin receptor (IR). In contrast to this, Ser-phosphorylation of IRS-1/-2 leads to a block in the IR-dependent induction of proliferative signals [9].

In light of the relevance of the Nrf2/ARE system for the process of liver regeneration and of the pathogenesis of an active chronic HBV infection that is associated with fibrosis and cirrhosis [278], it is obvious that the effect of HBV on the Nrf2/ARE system is of major interest. The literature describing this point is conflicting but this depends in part on the experimental systems that were used [119,279,280,281,282]. Therefore, several experimental settings have to be distinguished. Data obtained from HBV replication in cell culture in the absence of an immune response might differ from biopsy material of an acute or chronic active infection with a variety of effects triggered i.e., by the immune response. Moreover, analysis of HBV-associated HCCs reflects an endpoint and does not automatically represent processes affected on the way to the HCC.

Cell-culture data based on HBV-replicating cells generated either by transfection of overgenomic constructs or by infection of susceptible cells, provide evidence that HBV has the potential to induce an activation of NRF2/ARE-dependent gene expression. Further analyses revealed that the regulatory proteins HBx and LHBs (PreS2 activator) have the potential to activate the Nrf2/ARE-dependent gene expression [281,283]. This was observed based on reporter gene assays expressing a luciferase under the control of Nrf2-dependent minimal promoters, expression analyses using rtPCR for the quantification of transcripts of NQO1, GPx, GCLC or PSMB5 that all harbor Nrf2-dependent ARE sites in their promoter, and on Western blot analyses and immunofluorescence microscopy using GPx, GCLC, PSMB5 and NQO1-specific antisera. The elevated expression of these cytoprotective genes correlates with an enhanced capacity to detoxify ROS that disappears if a tdn (transdominant negative) mutant of Nrf2 is coexpressed [281,283,284].

Both regulatory proteins trigger via different initial steps an activation of c-Raf. Activation of c-Raf was described as being causative for the HBV-dependent activation of Nrf2 [281] (Figure 2). Inhibition of Nrf2 by a small molecule inhibitor (sorafenib) as well as by coexpression of a tdn mutant abolishes HBV-dependent activation of Nrf2. It was also reported that HBx triggers activation of Nrf2 by formation of a ternary complex consisting of p62, HBx and Keap1 [283].

HBx was described to interact with the outer mitochondrial membrane and thereby to destroy the membrane integrity of the outer mitochondrial membrane. Moreover, HBx binds to VDAC3 and thereby affects the permeability transition pore (PTP). HBx was described to increase the Ca^2+^ concentration in the cytoplasm and mitochondria that affects the PTP function. In addition, HBx decreases the activity of the respiratory chain complex by an inhibitory effect on the expression of subunits of the respiratory complexes I, II, IV and V. All these steps can finally cause a breakdown of the mitochondrial membrane potential and thereby lead to ROS production.

Overexpression of LHBs leads to retention and accumulation in the ER that is associated with ER stress and induction of the UPR. The UPR leads to a release of Ca^2+^ in the cytoplasm that can trigger the induction of ROS.

HBx and LHBs (if the PreS2-domain faces the cytoplasm) are transcriptional activators. Both regulatory proteins cause an activation of c-Raf that was described as crucial for the HBV-dependent activation of Nrf2. The HBV-dependent activation of Nrf2 triggers the expression of cytoprotective genes harboring ARE sequence(s) in their promoter. The increased expression of cytoprotective genes in HBV expressing cells enables the detoxification of ROS.

Moreover, there are conflicting data about the expression of NQO1 in HBV replicating cells. In contrast to the described HBV-dependent induction of the NQO1 expression [281] there are reports that HBx has the capacity to suppress the expression of NQO1 and of MTH1/MTH2 due hypermethylation of its promoter [285,286]. The HBx-mediated recruitment of DNMTA3 methyltransferase can lead to hypermethylation of the respective promoter [287,288]. Moreover, there are reports about a HBX-dependent repression of GSTM3 (glutathione-S-transferase M3) [289] and GSTP1 (glutathione-S-transferase P1). An interesting aspect in this context is that the HBV-dependent repression of GSTP1 expression seems to depend on the HBV genotype. While HBV genotype D was described to suppress the expression this was not found for the genotypes A to C [280]. This fits to recent observations that the HBV genotypes differ with respect to their potential to modulate the Nrf2/ARE-dependent gene expression. Cell-culture experiments provided evidence that HBV genotype D leads to a significantly weaker induction of the Nrf2/ARE-dependent gene expression as compared to genotype A or genotype G compared to A [284]. It is tempting to speculate whether differences in the Nrf2/ARE-dependent gene expression have an impact on the differences between the HBV genotypes regarding their tendency to establish chronic infections and with respect to the virus-associated pathogenesis.

## 11. HBV Regulatory Proteins

A variety of functions is ascribed to HBx. While there is an activating effect of HBV on the Nrf2/ARE system there is no evidence that activation/ inhibition of Nrf2 has a direct effect on HBV replication as evidenced by coexpression of constitutive active (ca) or tdn mutants in HBV-replicating cells. However, in the context of a natural infection leading to an immune response the HBV-dependent activation of Nrf2 could represent a viral strategy to escape from the immune response [281]. In the context of a cellular immune response, induction of ROS is frequently found in order to contribute to the elimination of the infected cell and to suppress viral replication. The Nrf2/ARE-dependent induction of cytoprotective genes leads to and inactivation of ROS and thereby counteracts the ROS-dependent effect [37,278].

Among the Nrf2-ARE-regulated genes there are in addition catalytical active subunits of the constitutive proteasome i.e., PSMB5 and PSMB3 [37,39]. In accordance with this, an elevated level of the activity of the constitutive proteasome was found in HBV-expressing cells as compared to the negative control. As the elevated activity of the constitutive proteasome was found to be associated with a decreased activity of the immunoproeasome, it was speculated that this might represent an additional escape strategy from the cellular immune response. It has been observed that the elevated Nrf2-activity leads to an elevated activity of the constitutive proteasome that is associated with a decreased activity of the immunoproteasome [281]. Based on this, it is hypothesized that the decreased activity of the immunoproteasomal system leads to a reduced processing of HBV-specific antigens and subsequent presentation of HBV-specific peptides that finally leads to a reduced cellular immune response. The experimental proof for this hypothesis, however, is still open.

Apart from the effect on the Nrf2/ARE system, HBV affects further antioxidant defense systems. Among these are two GST isoforms (GSTO1 and GSTK1) that are not Nrf2/ARE-dependent regulated. For both isoenzymes an elevated expression is found in HBV-replicating cells [290]. In addition, this is observed for peroxiredoxin 2 in patients suffering from chronic HBV infection. There are conflicting data about the effect of HBV on SOD2 (superoxide dismutase) that could depend on the chosen experimental systems [282,291,292].

## 12. Effect of ROS on the HBV Life-Cycle 

There is a recent report that described that H_2_O_2_ leads to an enhanced replication of HBV. Interestingly, there is in addition a promoting effect of H_2_O_2_ on the capsid assembly. It is suggested that H_2_O_2_ favors the formation of core Hsp90 complexes that support capsid assembly [293].

NAC (N-acetylcysteine) was described to exert an inhibitory effect on HBV. The inhibitory effect of NAC on HBV replication could be due to an indirect effect. NAC prevents the proper formation of disulfide bond formation in the HBsAg and thereby leads to an impaired release of subviral and viral particles [294,295].

## 13. ROS, Nrf2 and the Virus-Associated Pathogenesis

Although there is evidence that HBV per se does not directly lead to an elevated level of ROS, there is evidence that in many cases in patients suffering from chronic HBV infection, an elevated ROS level can be found as evidenced i.e., by quantification of the 8-OH dG level in liver samples [296,297,298]. Elevated ROS levels in patients suffering from acute or chronic HBV infection might be a multifactorial process. As described HBV i.e., HBx has the capacity to induce the formation of ROS [119,299]. However, in principle this could be compensated for by the induction of cytoprotective mechansims. In light of this, elevated ROS levels in HBV-positive patients could reflect an impaired induction of cytoprotective mechanisms or—more likely—an overload of the detoxifying system. Apart from the direct HBV-dependent induction of ROS i.e by HBx, there are indirect mechanisms. Both regulatory proteins of HBV (HBx and LHBs) have the capacity to activate NF-kB [268,300,301,302]. On the one hand, the HBV-dependent activation of NF-kB leads to the induction of proinflammatory cytokines like TNFα, lymphotoxin-α or IL-6 and on the other hand HBx was described to suppress the expression of anti-inflammatory cytokines [302,303,304,305,306]. Major sources of ROS are infiltrating NKs and CTLs. In case of a chronic HBV infection that is frequently characterized by an inefficient cellular immune response, there is a permanent but insufficient immune response that might significantly contribute to the formation of ROS.

Elevated ROS level are a major factor for the development of liver carcinogenesis. ROS affect the integrity of the genomic and of the mitochondrial DNA. In accordance to this, in HBV positive tissues an elevated level of 8-OH dG can be found and on the one hand an increased expression of OGG1 (8-oxoguanine glycosylase 1), a DNA repair enzyme [296,299,307,308]. On the other hand, a decreased expression of APE-1 is described to be associated with HBV infection [309]. Moreover, there is evidence that oxidative damage of DNA in HBV-infected cells is associated with DNA single strand breaks [310,311]. In accordance to this, an activation of the ATM-Chk2 pathway in HBV-infected cells that is involved in the repair of DNA strand breaks was described [312].

According to the classic two-step model of carcinogenesis that encompasses two steps (initiation and promotion) [313,314] elevated ROS level could fulfill the function of an initiator leading to an accumulation of critical mutations in the DNA. Expression of the regulatory proteins could act in a tumor promoter-like function [269,315,316] by activation of pathways that mediate a positive selection/ growth advantage of these cells.

Moreover, induction of strand breaks can be a factor mediating the integration of viral DNA into the host genome [310,317,318]. The effect of the various HBV genotypes on the ROS level and the impact of the ROS level on formation of DNA integrates is still unclear.

Almost all HBV-associated HCCs harbor chromosomally integrated DNA [319,320]. Due to the circular structure of the HBV genome, no formation of infectious viral particles can occur based on the integrated DNA, but the expression of regulatory proteins occurs. Moreover, there are reports about insertion of HBV-DNA thereby affecting the expression/ function of key enzymes controlling cell-cycle and proliferation [321,322,323,324,325]. Although there are interesting integrates identified and characterized, there is no evidence that deregulation of cell-cycle control by integration of HBV-specific DNA is a general phenomenon. In contrast to this, in case of WHV infection integration of WHV DNA into the *c-Myc* gene can be frequently found supporting the cis-hypothesis for the WHV system [326]. In case of HBV infection there is no preferred insertion into the *c-Myc* region observed, although a recent study observed that in 12.4% of early-onset HCCs an integration of *c-Myc* and PVT-1 occurs [327].

## 14. Liver Regeneration

It is an established model that elevated ROS levels lead to insulin resistance [328]. In light of the prominent role of insulin/IR-dependent signaling for the process of liver regeneration, elevated ROS levels are considered as a major factor contributing to impaired liver regeneration and fibrosis/cirrhosis induction.

However, in in vitro cell culture models of HBV-replicating cells, insulin resistance was observed although no elevated level of ROS was found [276]. Further analyses revealed that in HBV-replicating cells insulin resistance can be mediated by a novel mechanism. In HBV positive cells an elevated expression and formation of α-taxilin is found [329]. α-taxilin initially was described as a syntaxin 4 binding protein [330,331]. However, α-taxilin exclusively binds to free syntaxin 4 that is not part of the SNARE-complex. Thus, α-taxilin acts as a negative regulator of SNARE-complex formation and thereby can act as an inhibitor of intracellular transport pathways. Indeed, overexpression of α-taxilin leads to an impaired transport of the IR to the cell surface and thereby uncouples the cell from the insulin signaling that finally contributes to an impaired liver regeneration process [276].

Many data characterizing the effect of HBV on the Nrf2/ARE system are based on the analysis of HBV-associated HCCs. It should be considered that the HBV-associated HCC stands at the end of a long process starting with the acute infection. A variety of factors contributes to the formation of an HCC. Interpreting data based on the analysis of HBV-associated HCC, it should be considered that the tumor represents a specific physiologic situations and that the positive selection of the tumor tissue can be due to the activation of escape strategies and cytoprotective mechanisms of the tumor that favor the growth of the tumor and do not automatically reflect the impact of HBV on the healthy tissue.

A further critical point that should be considered are the data that are based on selective overexpression of the regulatory proteins. Especially HBx is produced in small amounts during the natural infection process, as evidenced for WHV—a situation that differs from the strong overexpression systems [332,333]. To consider this is of special importance for the evaluation of functional data that are based on the formation of stoichiometric complexes of HBx with abundant cellular proteins. Due to the low amount of HBx the general physiological relevance of these data has to be interpreted carefully.

Finally, we learn that the HBV genotypes differ with respect to their geographic distribution [334], to their molecular virology and to their associated pathogenesis. In light of this, it will be challenging to establish a closer correlation between these different factors that might strongly deepen our understanding of virus–host interaction and, thereby, could contribute to the development of novel prognostic markers and therapeutic tools.

## 15. Conclusions

Apart from the effect of the cellular immune response on the intracellular ROS level in infected cells, both viruses strongly differ with respect to their effect on radical generating and detoxifying systems. For HCV-positive cells there are a variety of reports describing the stimulating effect of HCV on radical producing systems on the one hand and an inhibitory effect on detoxifying systems as the Nrf2/ARE-dependent gene expression. The elevated ROS level can be correlated with insulin resistance that finally contributes to the development of fibrosis, cirrhosis and HCC. With respect to the viral life cycle the elevated ROS level is involved in the induction of autophagy that is crucial for the HCV life cycle. An open question is whether the elevated ROS level is a causative factor for the high genetic variability of HCV by affecting the integrity of the RNA genome.

In contrast to this, the situation for HBV seems to be even more complex. On the one hand there is the effect of the cellular immune response and of the viral regulatory proteins that have the potential i.e., by interfering with the mitochondrial integrity to increase the intracellular ROS level. On the other hand, however, there are reports describing the activation of the Nrf2/ARE-dependent gene expression and increased production of cytoprotective enzymes in HBV-positive cells. The activation of the Nrf2/ARE system could represent an escape strategy i.e., by combatting ROS that are produced as part of the immune response to eliminate HBV-positive cells. There might be an equilibrium between ROS-inducing and ROS-inactivating mechanisms that changes over the long period from an acute to chronic infection and HCC development. At some steps an increased Nrf2-activation could impair the virus elimination at other steps an elevated Nrf2 activity could prevent the elimination of an HBV-associated HCC. Regarding this point, there are still a variety of open questions. Moreover, it turns out that the HBV genotypes differ with respect to the activity of their regulatory proteins and the virus-associated pathogenesis. It might be interesting to investigate the relation between HBV- genotype, impact on the ROS level and pathogenesis.

## Figures and Tables

**Figure 1 ijms-20-04659-f001:**
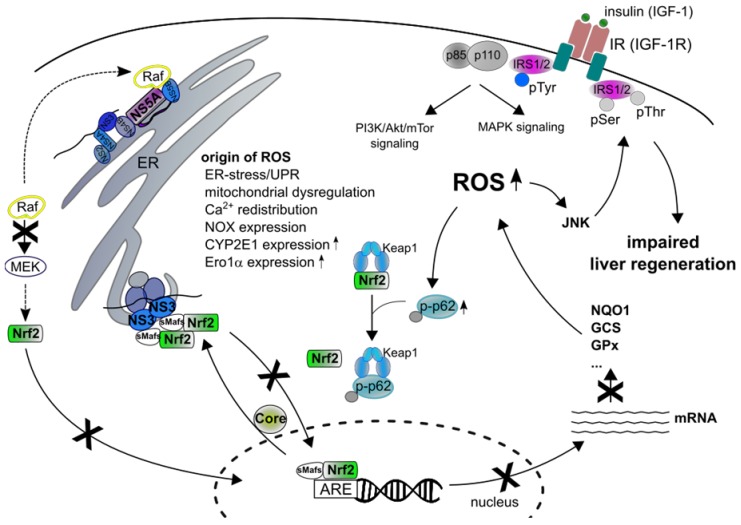
Interference of hepatitis C virus (HCV) with the nuclear factor erythroid 2 (NF-E2)-related factor 2 (Nrf2)/ Keap1-signaling pathway. Lines labelled with an “X” represent blocked processes, dotted lines describe indirect processes, solid lines describe translocations or direct effects.HCV is associated with oxidative stress in liver cells that results in increased levels of ROS. ROS in HCV infected hepatocytes can be related to ER-stress and the UPR, mitochondrial dysregulation, Ca^2+^ redistribution, activation of NADPH oxidases and enhanced expression of CYP2E1 and Ero1α. The increased ROS-levels further trigger phosphorylation of p62 on Ser349 that activates Nrf2. Based on the impaired Nrf2/Keap1-signaling pathway in HCV-infected cells, the oxidative stress cannot be compensated. This mechanism is based on a core-mediated delocalization of sMaf-proteins from the nucleus to the ER-associated RCs, where they are trapped by binding to NS3. As a consequence, the delocalized sMaf-proteins bind to Nrf2 and prevent the translocation of Nrf2 into to the nucleus resulting in impaired expression of cytoprotective genes. In addition NS5A interferes with Nrf2-activation via a crosstalk with the MAPK-signaling cascade. NS5A recruits cRaf to the ER-associated RCs resulting in activation of cRaf and NS5A phosphorylation. However, despite cRaf activation no activation of the MAPK signaling cascade could be observed resulting in impaired Nrf2-activation. The elevated ROS-levels further interfere with mechanisms involved in liver regeneration. Impaired Nrf2 activation results in a decreased tyrosine phosphorylation and enhanced serine/threonine-phosphorylation of IRS-1 and -2 that may contribute to the development of insulin resistance and impaired liver regeneration. However, the mechanisms regulating increased ROS-levels in HCV-infected cells are conflicting. UPR, unfolded protein response; NOX, NADPH oxidase; CYP2E1, cytochrome P450 E1; Ero1α, ER oxidoreductin 1α; MAPK, mitogen-activated protein kinase; IRS1/2, insulin receptor substrate; IR, insulin receptor; IGF-1, insulin-like growth factor 1; IGF-1R, insulin-like growth factor 1 receptor; JNK, c-Jun-N-terminal kinase; NQO1, NAD(P)H:quinone oxidoreductase 1; GPx, glutathione peroxidase; γ-GCS, γ-glutamylcysteine synthetase.

**Figure 2 ijms-20-04659-f002:**
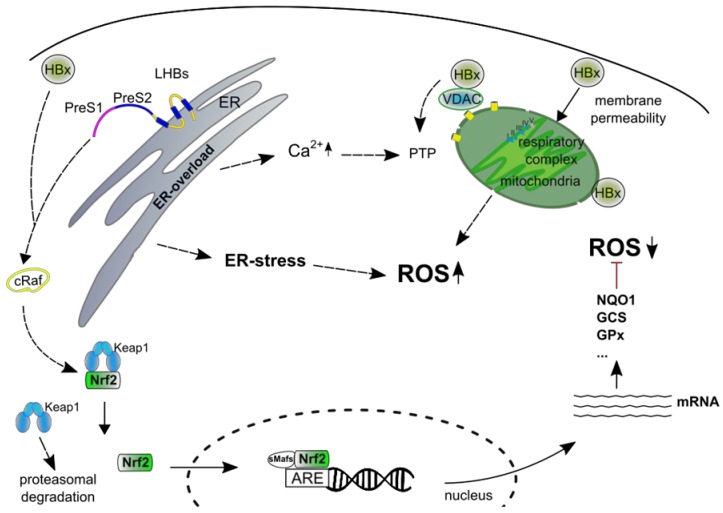
Effect of hepatitis B virus (HBV) on reactive oxygen species (ROS) production and inactivation. Dotted arrows describe indirect processes, solid lines describe translocations or direct effects.

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
