# Peer review of "Effect of Hepatitis Viruses on the Nrf2/Keap1-Signaling Pathway and Its Impact on Viral Replication and Pathogenesis"

_ijms, 2019, doi:10.3390/ijms20184659_

Round 1

Reviewer 1 Report

Very clear and exhaustive description concerning the Nrf2 pathway involvement in viral hepatitis.

First of all, I found that the description of the Nrf2 pathway regarding its regulation and its importance looking at the new concept of Systems Medicine has been reported clearly and exhaustively in all the different epigenetic and translational mechanisms.

The fact that this pathway is involved in hepatitis C and B could open new perspectives and therapeutic potentials regarding the best approach in treating the patients affected by these pathologies.

Author Response

Thank you for the encouraging review.

Reviewer 2 Report

The manuscript is properly written and well-presented about the effect of hepatitis virus infection on Nrf2 signaling pathway. The reviewer thinks the manuscript is acceptable for IJMS with revisions.

1) T-shaped red lines in Figure 1 would be confusing for readers, since T-shaped line usually used for “inhibition”, not for “down regulation” as shown in Figure 2.

2) Page 3, line 114-117 The sentence (Moreover…..response) looks inappropriate description for this section.

In addition, Nrf2/NQO1 and HO-1 axis rather suppresses inflammation.

3) There are some typographical errors.

Page 2, line 50 &51, cm3 (use superscript)

Page 2, line 75, insert "sites" after “AP-1 (activating protein 1)".

Page 2, line 75, "promotor" should be "promoter".

Page 2, line 85, It would be better to use "domain structure" rather than "structural elements".

Page 2, line 88, “sMafK” should be “MafK”. sMaf means "small Maf".

Page 3, line 109, “aryl hydrogen carbonate” should be “aryl hydrocarbon”.

Page 3, line 112, “miR” is microRNA?

Page 3, line 119, “prostata” should be “prostate”

Page 3, line 128 & 133, “Cul3-RBX” should be “Cul3-RBX1”.

Page 4, line 170, “XBP” should be “XBP1”.

Page 4, line 178, “sequestome 1” should be “sequestosome 1”

Page 3, line 172, Ref. 84 is correct?

Page 5 line 194, Ref 82 is correct?

Page 5, line 194, “autopagosomal” should be “autophagosomal”.

Page 6, line 260, “hydrogenperoxide” should be “hydrogen peroxide”. Page 7, line294, 295, 305 also.

Page 7, line 330, “Carvajal” should be “Carvajal-Yepes”.

Page 10, line 477, [227-230] Ref 230 is correct?

Page 11, line 494, “p61” should be “p62”.

Page 11, line 510, “insulin growth factor” should be “insulin-like growth factor”.

Page 15, line 691, “acetylcystein” should be “acetylcysteine”

Page 16, line 698 & line 714, “8OH-dG” should be “8-OH dG”.

Page 18, There are some error in abbreviations (AhR, AMPK etc.)

Author Response

Point by point reply to the comments of reviewer 2:

Reviewer:” T-shaped red lines in Figure 1 would….. “

The figures 1 and 2 were changed as suggested by the reviewer Reviewer: “ Page 3, line 114-117 The sentence (Moreover…..response) looks inappropriate ….”

In the revised version we deleted this sentence to avoid confusion. Reviewer: “ There are some typographical errors….”. We apologize for the typing errors and are grateful to the reviewer for the careful review. All typing errors listed by the reviewer were corrected, the Thank you! All typing errors and the references in doubt were corrected as requested by the reviewer.

Page 2, line 50 &51, cm3 (use superscript)

corrected

Page 2, line 75, insert "sites" after “AP-1 (activating protein 1)".

corrected

Page 2, line 75, "promotor" should be "promoter".

corrected

Page 2, line 85, It would be better to use "domain structure" rather than "structural elements".

corrected

Page 2, line 88, “sMafK” should be “MafK”. sMaf means "small Maf".

corrected

Page 3, line 109, “aryl hydrogen carbonate” should be “aryl hydrocarbon”.

corrected

Page 3, line 112, “miR” is microRNA?

corrected

Page 3, line 119, “prostata” should be “prostate”

corrected

Page 3, line 128 & 133, “Cul3-RBX” should be “Cul3-RBX1”.

corrected

Page 4, line 170, “XBP” should be “XBP1”.

corrected

Page 4, line 178, “sequestome 1” should be “sequestosome 1”

corrected

Page 3, line 172, Ref. 84 is correct?

corrected

Page 5 line 194, Ref 82 is correct?

corrected

Page 5, line 194, “autopagosomal” should be “autophagosomal”.

corrected

Page 6, line 260, “hydrogenperoxide” should be “hydrogen peroxide”. Page 7, line294, 295, 305 also.

corrected

Page 7, line 330, “Carvajal” should be “Carvajal-Yepes”.

corrected

Page 10, line 477, [227-230] Ref 230 is correct?

corrected

Page 11, line 494, “p61” should be “p62”.

corrected

Page 11, line 510, “insulin growth factor” should be “insulin-like growth factor”.

corrected

Page 15, line 691, “acetylcystein” should be “acetylcysteine

corrected

Page 16, line 698 & line 714, “8OH-dG” should be “8-OH dG”.

corrected

Page 18, There are some error in abbreviations (AhR, AMPK etc.)

corrected